# News Video Summarization Combining SURF and Color Histogram Features

**DOI:** 10.3390/e23080982

**Published:** 2021-07-30

**Authors:** Buyun Liang, Na Li, Zheng He, Zhongyuan Wang, Youming Fu, Tao Lu

**Affiliations:** 1School of Computer Science, Wuhan University, Wuhan 430072, China; liangbuyun@163.com (B.L.); wzy_hope@163.com (Z.W.); fuym@whu.edu.cn (Y.F.); 2The Archives of Wuhan University, Wuhan University, Wuhan 430072, China; 3School of Computer Science and Engineering, Wuhan Institute of Technology, Wuhan 430073, China; lut@wit.edu.cn

**Keywords:** video summarization, SURF features, clustering, shot boundary detection, key frame extraction

## Abstract

Because the data volume of news videos is increasing exponentially, a way to quickly browse a sketch of the video is important in various applications, such as news media, archives and publicity. This paper proposes a news video summarization method based on SURF features and an improved clustering algorithm, to overcome the defects in existing algorithms that fail to account for changes in shot complexity. Firstly, we extracted SURF features from the video sequences and matched the features between adjacent frames, and then detected the abrupt and gradual boundaries of the shot by calculating similarity scores between adjacent frames with the help of double thresholds. Secondly, we used an improved clustering algorithm to cluster the color histogram of the video frames within the shot, which merged the smaller clusters and then selected the frame closest to the cluster center as the key frame. The experimental results on both the public and self-built datasets show the superiority of our method over the alternatives in terms of accuracy and speed. Additionally, the extracted key frames demonstrate low redundancy and can credibly represent a sketch of news videos.

## 1. Introduction

News video summarization tasks aim to extract the key frame sequence from a complete and long news video to summarize the news video, to meet the needs of users for quickly browsing and understanding the content [1]. In video summarization, a key frame extraction algorithm is a feasible and effective method. This method summarizes the video by splitting it into individual shots and then extracting key frames from each shot.

In shot segmentation, the mainstream methods extract the features of video frames and judge whether they are located on the boundary of the shots by comparing the differences between the two frames. Wu et al. [2] introduced a method to directly compare pixel differences between two frames, but the method was easily affected by the motion and rotation of the shots. Yang et al. [3] used Hu-invariant-moment to extract features for initial inspection, and then carried out re-inspection through color features to determine the boundary of the shots. Bae et al. [4] realized shot boundary detection through discrete cosine transform and pattern matching of color histograms after extracting the color features of the frame. Zheng et al. [5] improved the efficiency of color feature extraction by using the MapReduce platform. However, the disadvantages of the above methods using color features are that two completely different frames may also show similar color features, thus causing false detection. Tang [6] compared the difference between frames by extracting the ORB (oriented fast and rotated BRIEF, BRIEF: binary robust independent elementary features) features in video frames, in order to detect shot boundaries. Despite the high processing speed of ORB features, it can easily result in false detection when the shot is zooming. Rachida et al. [7] extracted SIFT feature points as local features of frames to construct a SIFT–PDH histogram, and then defined the distance of SIFT–PDH between frames and selected an adaptive double threshold to realize the detection of shot boundaries. SIFT features can maintain a high stability against such factors as the rotation and motion of the shots, the diversity of object sizes, the intensity of light and changes in brightness. However, due to the large amount of calculation in the SIFT algorithm, the real-time performance is restrained. Bendraou et al. [8] used a low-rank matrix approximation of Frobenius norm to extract intra-frame features. In recent years, deep learning technology has been used to learn the inherent characteristics of data from a large number of samples and is particularly suitable for visual computing tasks [9,10,11,12], including video detection, classification and segmentation. Zeng et al. [13] extracted the features of frames and calculated the similarity between frames by recurrent neural network (RNN), to realize the detection of abrupt and gradual frames. However, this method neglects the applicable scope of the features and cannot express the frame information well.

In the research of key frame extraction, Zhong et al. [14] realized key frame extraction by building a fully connected graph. Qu et al. [15] proposed an extraction algorithm based on SIFT feature points, but the extraction algorithm based on local features showed high redundancy and low real-time performance. Qu et al. [16] selected the frame with the maximum image entropy as the key frame from each shot. Liang et al. [17] extracted the deep features of the frame by constructing a convolutional neural network (CNN), and then selected the frame containing the most significant feature as the key frame. The above two methods can only select one key frame in a shot. Clustering algorithms are also widely employed in key frame extraction. Sun [18] and Sandra [19] used the K-means clustering algorithm to determine the location of key frame. The major disadvantage of the K-means clustering algorithm is that the number of clusters needs to be set in advance, which means it cannot well adapt to changes in shot complexity.

For video summarization, many types of research have been carried out in the past and are ongoing until now. Souček et al. [20] presented an effective deep network architecture TransNetV2 for shot detection, which provided promising detection accuracy and enabled efficient processing of larger datasets. Liu et al. [21] proposed a hierarchical visual model, which hypothesized a number of windows possibly containing the object of interest. Zhang et al. [22] proposed a context-aware video summarization (CAVS) framework, where sparse coding with a generalized sparse group lasso was used to learn a dictionary of video features as well as a dictionary of spatio-temporal feature correlation graphs. Jurandy et al. [23] exploited visual features extracted from the video stream and presented a simple and fast algorithm to summarize the video content. Recently, Workie et al. [24] provided a comprehensive survey on digital video summarization techniques, from classical computer vision until the recent deep learning approaches. These techniques fall into summarized, unsupervised and deep reinforcement learning approaches. Cahuina et al. [25] presented a method for static video summarization using local descriptors and video temporal segmentation. It uses a clustering called “X-mean” and needs to specify two parameters in advance to determine the number of clusters, leading to inconvenience in practice. Therefore, video summarization still faces various challenges, including computing equipment, complexity and lack of datasets.

In order to overcome the shortcomings of the above algorithms, this paper proposes a shot boundary detection algorithm based on SURF features [26] and a key frame extraction algorithm with an improved clustering algorithm for summarizing news videos. First, by regarding SURF features as local features of frames, this algorithm extracts SURF feature points from each frame and matches them. The similarity between adjacent frames is calculated according to the matching results and then a similarity curve between frames is depicted. Based on the similarity curve, the mutation (sudden switch) and gradient (gradual switch) of the shots are detected by selecting the double thresholds. Second, for each shot, an HSV color histogram of the frame in the shot is extracted, and then the number of clusters is dynamically determined by clustering the color histograms through the improved clustering algorithm. After the clustering is completed, the frame closest to the center is selected as the key frame in each cluster. SURF features can not only maintain invariability for shot rotation and scale variations, but also enjoy good stability in shot motion, noise and brightness changes. Because the proposed clustering algorithm does not need to set the number of clusters in advance, the number of extracted key frames is able to agree with the complexity of shots.

In brief, the major contributions of this paper are highlighted as follows.

(1) In order to facilitate SURF based shot detection, we propose an inter-frame similarity metric based on the SURF feature points respectively extracted from two frames and the matched SURF feature points as well. Furthermore, in order to improve the detection accuracy of abrupt and gradual shots, we propose a double-threshold method to locate the shot boundary in the similarity curve.

(2) We propose an improved HSV color histogram clustering algorithm to extract the key frame within a shot. The algorithm can dynamically determine the number of cluster centers without pre-setting in advance, so it can adapt to variations in the complexity of shots.

The remainder of this paper is organized as follows. In Section 2, we introduce the SUFR description algorithm. In Section 3, the proposed video summarization method is detailed. Experimental results are given and analyzed in Section 4. Section 5 concludes the paper.

## 2. Preliminaries

The speeded up robust feature (SURF) [26] local feature description algorithm can maintain high stability for shot rotation, shot motion, object size change, light intensity and brightness change. The SURF algorithm improves on the SIFT algorithm through some steps, making the algorithm faster. The SURF algorithm includes three main steps: detecting the feature points, principal direction determination and feature descriptor generation.

(1)Feature point detection

The SURF algorithm detects feature points by calculating the determinant of the Hessian matrix of images at different scales. A Hessian matrix is a second order partial derivative matrix. For the point x=(x, y) in a given image, the Hessian matrix H(x,σ) defined at the point x with the scale σ of is shown in Equation (1).
(1)$$H\left(x,\sigma \right)=\left[\begin{array}{cc} L_{xx}\left(x,\sigma \right) & L_{xy}\left(x,\sigma \right)\\ L_{xy}\left(x,\sigma \right) & L_{yy}\left(x,\sigma \right) \end{array}\right]$$
where $L_{xx}\left(x,\sigma \right)$, $L_{xy}\left(x,\sigma \right)$ and $L_{yy}\left(x,\sigma \right)$ are Gaussian second order partial derivatives. The Hessian matrix of the image reflects the local curvature of images after Gaussian filtering.

The SURF algorithm uses box filters to approximate the Gaussian second partial derivatives. Figure 1 compares the box filter with the Gaussian second derivative template. As seen, the box filters $D_{xx},\;D_{xy}$ and $D_{yy}$ approximately replace the Gaussian second partial derivatives $L_{xx}$, $L_{xy}$ and $L_{yy}$, improving the convolution efficiency of images.

The Hessian matrix constructed by the SURF algorithm and the determinant of the Hessian matrix [26] are defined as follows:(2)$$H_{SURF}=\left[\begin{array}{cc} D_{xx}\left(x,\sigma \right) & D_{xy}\left(x,\sigma \right)\\ D_{xy}\left(x,\sigma \right) & D_{yy}\left(x,\sigma \right) \end{array}\right]\;$$
(3)$$Det\left(H\right)=D_{xx}D_{yy}-{\left(0.912D_{XY}\right)}^{2}\;$$
According to the determinant of the Hessian matrix, whether the pixel is an extreme point can be determined. When the determinant is positive, it means that the pixel is an extreme point under the scale σ.

(2)Determination of the main direction

In order to make SURF feature points have rotation invariance, SURF determines a principal direction for each feature point. In the SURF algorithm, a circular region with feature points as the center and 6 s (s refers to the corresponding scale of feature points) as the radius is first established. Then, a 4 s-sized Haar wavelet template is used to scan the horizontal and vertical directions of the region, and the horizontal and vertical Haar wavelet responses are calculated. Finally, it scans the 60-degree sector window in this area to select the direction of the maximum total horizontal and vertical responses of Haar as the main direction of the feature point.

(3)Generation of feature descriptors

A square box with side length of 20 s and direction as the main direction of the feature point is constructed at the feature point. In addition, the window is divided into 4 × 4 sub-boxes, where each box has a side length of 5 s. In each box, statistics of the horizontal Haar wavelet response, dx, the vertical direction of the sum of Haar wavelet response, dy, the absolute value of horizontal direction in the sum of Haar wavelet response |dx| and the absolute value of vertical direction Haar wavelet response of the sum |dy| are combined. Thereby, a four-dimensional vector is formed in each region. The four-dimensional vectors formed in each region are expressed as follows:(4) V=[∑dx,∑dy,∑|dx|,∑|dy|]

## 3. News Video Summarization

The video summarization method proposed in this paper can be divided into two parts. First, the whole news video is divided into individual video clips using the shot boundary detection based on SURF features. Then, the improved clustering algorithm is used to realize the key frame extraction in each shot. Finally, a summary of the news video is generated by arranging key frames chronologically. Note that, in this paper, key frames refer to the representative frames that can reflect the outline of the video contents.

### 3.1. SURF Based Shot Boundary Detection

The SURF based shot boundary detection algorithm includes four steps: video preprocessing, SURF feature point extraction and matching, inter-frame similarity calculation, double thresholds selection and shot boundary detection.

#### 3.1.1. Video Preprocessing

Because the main sources of news videos are TV programs, and news videos from TV usually have the platform logo and a scrolling subtitle at the bottom, the logo and the scrolling subtitle usually do not change when the video shot changes. Therefore, the presence of the platform logo and scrolling subtitles will affect the detection of the shot boundary. In order to improve the accuracy of detection, it is necessary to remove the logo and the scrolling subtitle in the news video frame first.

The logo usually stays the same in the video, so we find the logo by looking for pixels that do not change or change very little in the video. For the removal of the logo, we first evenly select a number of frames $f_{1,2,3\ldots n}$, and then, based on these frames, a simple rule is used to identify the logo. We use a map to indicate unchanged pixels, and for the output map *M*, the following equation is satisfied:(5)$$\;M\left(x,y\right)=\{\begin{array}{c} \left(0,0,0\right)\;if\;f_{1}\left(x,y\right)=f_{2}\left(x,y\right)\ldots =f_{n}\left(x,y\right)\;\\ \left(1,1,1\right)\;else \end{array}$$
where *x* and *y* represent the coordinates of the pixel. The three-dimensional vector (r, g, b) indicates the three color channels of R, G, and B, respectively.

After that, we use the original frames and the map *M* to perform the Hadamard product, so that we can get the image with the logo removed by
(6)$$I_{after}=I_{\mathrm{pre}}⊙M$$
⊙ refers to the Hadamard product, $I_{\mathrm{pre}}$ represents the original frame, and $I_{after}$ denotes the frame without the logo. According to the characteristics of TV news, we cut out the bottom 15% of the video to remove the scrolling subtitle.

Figure 2 shows the video frame before and after processing. Through the pre-processing step, the success rate of shot boundary detection can be improved, while the computational burden of subsequent steps can be reduced to improve the processing efficiency.

#### 3.1.2. Extraction and Matching of SURF Feature Points

After video preprocessing, we need to extract the features of video frames. Since SURF features keep stable for shot rotation, scale transformation, occlusion, lighting, and noise, this paper adopts the SURF algorithm to extract features of video frames.

After extracting the feature points and descriptors of each frame, in order to measure the difference between frames, feature point matching between adjacent frames is used. Two feature matchers are provided in OpenCV, namely, the Brute-Force feature matcher and the FLANN feature matcher. In order to improve the processing efficiency of the algorithm, the FLANN matcher with its faster matching speed is used in this paper, where the K-D tree nearest neighbor algorithm is used to select match points. After the feature preliminary matching by FLANN, we further adopt the RANSAC algorithm to eliminate the feature mismatch. Figure 3 illustrates an example of SURF feature extraction and feature matching. It can be seen that the RANSAC algorithm does eliminate a mismatch in Figure 3a.

#### 3.1.3. Inter-Frame Similarity

After finishing the matching of SURF feature points of adjacent frames, we quantitatively measure inter-frame similarity to reflect the difference between the two frames. Since the matching result of the SURF features reflects the consistency of the two frames to a certain extent (the more feature points on the match, the more similar the two frames), we can exploit the SURF feature matching results to measure the similarity between video frames. Following this idea, the similarity score between frame *i* and frame *j* is computed by
(7)$$\;S_{ij}=\frac{C_{ij}}{\max\left(k_{i},k_{j}\right)}\;$$
where $S_{ij}$ is the inter-frame similarity of frame *i* and *j*, $C_{ij}$ is the number of matched SURF feature points between frame *i* and *j*. $k_{i}$ and $k_{j}$ are the number of SURF feature points of frame *i* and frame *j*, respectively. Then, the similarity between all the adjacent frames is calculated and the similarity curve is depicted. Figure 4 shows an example of the similarity between frames.

#### 3.1.4. Selection of Double Thresholds and Shot Detection

Figure 5 shows typical inter-frame similarity curves for the abrupt and gradual shots, respectively. When the shot mutates, the video goes straight from one shot to another without any transition. Therefore, when the shot mutates, the inter-frame similarity drops to an extremely low level. When the gradual shot transition occurs, the similarity between the frames will also decrease, but the amplitude is not as drastic as with a sudden change in the shot. At the same time, the shot gradient lasts for a long time, generally lasting more than 10 frames.

Therefore, according to the characteristics of the mutation and gradient of the shots, this paper describes a double-threshold method for the detection of the shot boundary. For the set two thresholds Threshold1 and Threshold2, the following rules are regulated.

(1) When the similarity of a frame is less than Threshold1, it is judged as an abrupt frame. When the similarity of successive frames is greater than Threshold1 but less than Threshold2, the first frame of these frames is the starting frame of gradient, and the last one is the ending frame of gradient.

(2) A shot frame starts from a mutation or a gradient ending frame, and ends at the next mutation or gradient ending frame.

### 3.2. Clustering Based Key Frame Extraction

After splitting the video into individual shots, the key frames need to be extracted further within each shot. We can select a representative frame from within the shot as the key frame using a clustering algorithm. The clustering based key frame extraction in shots mainly consists of four steps, calculation of an HSV color histogram, histogram clustering, merging small clusters and selection of the key frame.

#### 3.2.1. Color Histogram Clustering

Compared with other color spaces, HSV color space is more consistent with human visual perception. Therefore, HSV color features are used as the feature representation for key frame extraction.

After extracting HSV color histogram of each frame, we cluster histograms with the following clustering algorithm.


**Step 1**


For all frames in the video, in order to make the result unaffected by the number of pixels in the video frame, the color histogram is normalized by
(8)$$N_{k}=\frac{O_{k}}{\sum_{i}O_{i}}\;$$
where $N_{k}$ represents the *k*-th value in the newly generated HSV color normalized histogram and $O_{k}$ denotes the *k*-th value in the original HSV color histogram.


**Step 2**


The algorithm generates a new cluster that contains the first frame of the video, the center of which is the value of the HSV color normalized histogram of the first frame.


**Step 3**


For frames that are beyond the first frame of the video, it calculates the distance between the HSV color histogram of that frame and the histogram of all the existing clusters. The distance between histogram *i* and histogram *j* is defined as
(9)$$Distance_{ij}=\underset{m}{\sum }\overset{N}{\underset{k=1}{\sum }}\left|v_{i,m,k}-v_{j,m,k}\right|\times W_{m}$$
where, *m* represents the color component, namely hue (H), saturation (S) and value (V). $v_{i,m,k}$ is the *k*-th value of normalized histogram *i* in the *m* color component. *N* refers to the total series of histogram *i*. $W_{m}$ is the weight of color component *m* in calculating distance.


**Step 4**


Given a cluster, *C*, if the distance between the frame and the cluster *C*, is less than the preset threshold, it is judged that the frame belongs to cluster *C*, and the center of cluster *C* is then updated. The cluster center is calculated by
(10)$$\;N_{k}=\frac{C_{k}\times \left(S-1\right)+F_{K}}{S}$$
where, $N_{k}$ is the *k*-th value of the centroid histogram after updating, $C_{k}$ is the *k*-th value of the centroid histogram before updating, and $F_{K}$ is the *k*-th value of the color histogram of the newly added frames. *S* is the total number of frames contained in the cluster after adding the frame to cluster *C*.

If there is no cluster *C*, the frame belongs to a new cluster, and the center of the cluster is initialized to the HSV color normalized histogram value of the frame.


**Step 5**


It repeats steps 3–4 until all frames in the shot are processed.

#### 3.2.2. Merge of Small Clusters

After the clustering is completed, a number of clusters are dynamically generated, with each cluster containing multiple video frames of the shots. However, the number of video frames in each cluster is inconsistent. Some clusters have more video frames, while some clusters have fewer frames. If the number of video frames contained in a cluster is small, it indicates that the cluster appears in the shot for a short time and is not representative. If key frames are extracted directly from a small cluster without additional processing, the extracted key frames will not be representative. In order to avoid redundancy of key frames and to make the selected key frames representative, it is necessary to merge the smaller clusters into the larger ones. We employ the following strategy to merge small clusters.

(1) If the size of a cluster satisfies
(11)S<W×F
the cluster will be merged. where, *S* is the number of frames contained in the cluster, *W* is the weight, and *F* is the total number of frames of the shot.

(2) The histogram distance between the centers of this cluster and other clusters is calculated, and the cluster with the smallest distance is selected to perform the merge, and the center of the merged cluster is updated.

#### 3.2.3. Selection of Key Frames

Based on the above method, the distance between the normalized histogram of the HSV frame and the center of the cluster is calculated by Equation (10). Then, the frame with the smallest distance from the center is selected as the key frame of the cluster. Therefore, if there are *k* clusters, *k* key frames can be given. The key frames are arranged in chronological order of the video to generate the final video summarization.

## 4. Experimental Results

### 4.1. Datasets and Evaluation Metrics

In order to confirm the validity of the method, a self-built news video dataset and a public dataset TVSum50 [27] were used for testing. The self-built dataset contains 13 video clips, which are excerpted from three mainstream news programs in China, namely, “News in 30 min”, “CCTV News” and “Sports Express”. The ground truth of the self-built dataset was manually extracted frame by frame. TVSum50 is a widely used dataset containing 50 videos, whose shot-level importance scores were annotated via crowdsourcing. Table 1 shows the details of the datasets.

Recall and precision were used for evaluation metrics. Recall represents the ratio of the number of true positive shots to the total number of shots. Precision refers to the rate of the true shots that are detected.

### 4.2. Results and Analysis

#### 4.2.1. Running Time

In order to verify the processing efficiency of the algorithm, we calculated the running speed of video summarization in frames per second. The results are shown in Table 2. Table 3 compares the feature extraction speed with SIFT based algorithms by Li, S. [28] and Li, F. [29].

It can be seen from the results that the speed of our algorithm in SURF feature point extraction is 41.27 frames per second, which is 29.6% faster than Li S. [28] and 56.13% faster than Li F. [29]. Evidently, the SURF based algorithm has a speed advantage over SIFT based algorithms.

#### 4.2.2. Comparison of Shot Boundary Detection

In order to verify the effect of this algorithm on shot boundary detection, we tested the recall and precision under different types of videos, in comparison with the algorithms by Feng [30], Wang [31], TransNetV2 [20] and Rachida [7]. Among them, Feng [30], Wang [31] and Rachida [7] are based on handcrafted features, while TransNetV2 [20] uses deep learning. Table 4 shows the experimental results of our algorithm, and Table 5 shows the comparison results.

From Table 5, we can see that our method shows significant advantages in recall rate both on self-built datasets and the TVSum50 dataset, which is mainly due to the fact that our method takes into account the gradient of shots. The recall rate of our method is respectively 10.74%, 6.81%, 0.73% and 4.54% higher than those of Feng [30], Wang [31], Rachida [7] and TransNetV2 [20]. Due to the fact that Feng [30] and Wang [31] adopt the shot boundary detection algorithm based on global features such as color and gray scale, they cannot detect some shot switches caused by local feature changes, resulting in a low recall rate. Instead, our method, and that of Rachida [7], employ the detection method of extracting local features within the frames, which can effectively detect the mutation and gradient of shots and gives a high recall rate. A low recall rate indicates that some video clips (which should have been recalled) have been missed while the low precision index only means that the extracted shots may appear redundant. Therefore, relatively speaking, the recall rate is the more important indicator in video summarization tasks. From this perspective, our method turns out to be more practical, regardless of recall rate or efficiency.

#### 4.2.3. Effects of Thresholds on Video Summarization

Threshold will also affect the extraction of key frames. The higher the threshold is set, the easier the video frames will be assigned into an existing cluster, and the lower the similarity of frames within the cluster will be. On the contrary, the lower the threshold is set, the more difficult it is for a frame to be merged into an existing cluster, and the easier it is to generate a new cluster, which may cause redundancy of key frame extraction.

Figure 6 shows the video summary results of the clips of “News in 30 Minutes” under different thresholds. When Threshold<0.2, a large number of key frames are available, which indicates that there appears a large amount of redundancy. When Threshold=1.0, although the extracted key frames do not exhibit redundancy, they cannot fully and accurately describe all the representative information in the video. When 0.4<Threshold<0.8, the extracted key frames show no redundancy and can reasonably reflect the dominant information in the video.

#### 4.2.4. Subjective Evaluation

In order to qualitatively examine the performance of our method, we conduct a comparison with a video summary made manually by users. Due to the limited space, Figure 7 shows only part of the results. As seen, the key frames extracted by our algorithm are highly similar to those extracted by the users, thus confirming the accuracy and effectiveness of the algorithm in extracting key frames.

We then evaluated the video summarization results extracted by the algorithm, taking segment 2 of “News 30” as a concrete example. As shown in Figure 8, the key frames extracted by our method effectively represents the content of the video, so that we can roughly judge the content and key pictures of the news from the extracted video summary. At the same time, the redundancy between extracted video summaries is low, and relatively repetitive key frames are rarely seen. Therefore, from the subjective point of view, the proposed method gives appealing performance and accuracy.

### 4.3. Ablation Studies

**Effects of different local descriptors.** In order to investigate the effectiveness of different local descriptors on the results, we tested the feature extraction speed and quality of different local descriptors. The visual and numeric comparisons on news video datasets are shown in Figure 9 and Table 6. It can be seen that the SURF descriptor is significantly faster than the SIFT descriptor. In addition, we can see that image preprocessing can improve the extraction speed of SURF features. Although it can be seen from Table 6 that BRIEF [32] is faster than SURF, Figure 9 shows that there are fewer feature points detected by BRIEF, which thus cannot represent the image content fully. Moreover, BRIEF features do not have scale invariance and cannot handle zooming, narrowing and rotation of objects in the shots.

**Effects of different color spaces.** We compared the intra-class distances of the clips of “News in 30 Minutes” in different color spaces. In order to ensure fairness, we normalized the histograms before calculation, and we use Bhattacharyya distance to represent the histogram distance. As shown in Table 7, the HSV color space gives a lower intra-class distance, which means a better clustering effect. This is because HSV is an intuitive color space for users. When the video shot has varied lighting and brightness, the RGB color space has to change the values of three channels, while the HSV color space often does not need to change all the channels.

## 5. Conclusions

News video summarization is a process of rapid summarization of news videos. This paper proposes a method of news video summarization based on SURF features and a clustering algorithm. The method consists of two steps: shot segmentation and key frame extraction. First, we preprocess the news video according to its characteristics, extract the SURF feature of frames as local features, and match the SURF feature points between consecutive frames. Then, we calculate the similarity scores between adjacent frames based on the matched points and determine the shot boundaries with the proposed double-threshold method. After the segmentation of the shots, color histograms of each frame in the shot are dynamically clustered, where smaller clusters are merged, and the key frames are extracted from each cluster to create the summary of the news video. The experimental results on news video datasets show that our method achieves an average accuracy of 93.33% and a recall rate of 97.22% in shot boundary detection. Moreover, the extracted key frames show a low redundancy and are highly similar to the video summaries made manually by users.

## Figures and Tables

**Figure 1 entropy-23-00982-f001:**
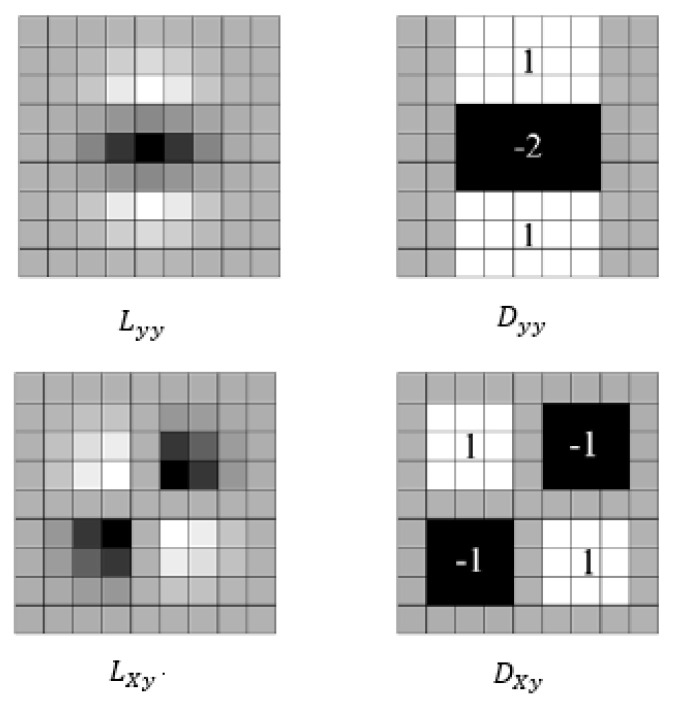
Gaussian second derivative templates.

**Figure 2 entropy-23-00982-f002:**
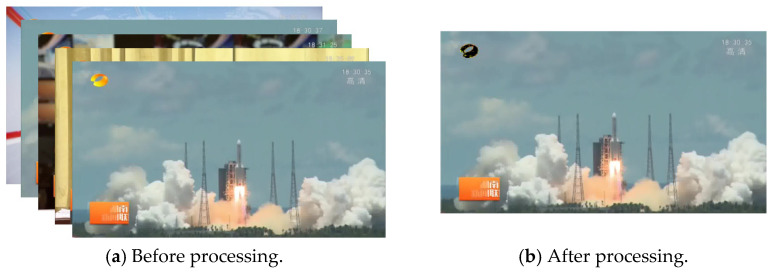
The results of video preprocessing.

**Figure 3 entropy-23-00982-f003:**
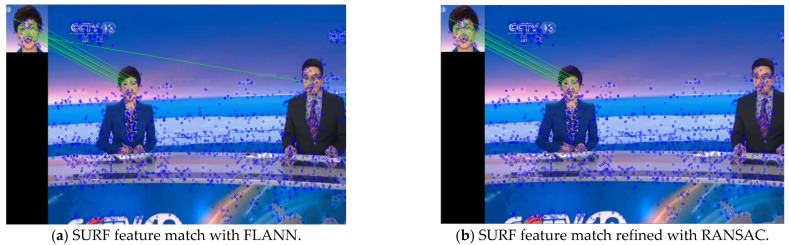
An example for extracting and matching SURF feature points.

**Figure 4 entropy-23-00982-f004:**
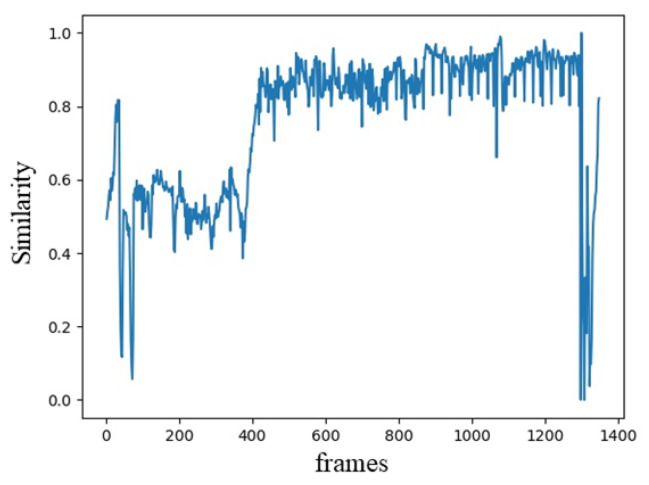
An example of inter-frame similarity curve.

**Figure 5 entropy-23-00982-f005:**
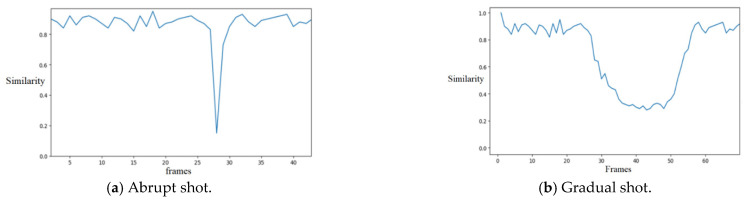
The similarity curves between frames for abrupt shot (**a**) and gradual shot (**b**).

**Figure 6 entropy-23-00982-f006:**
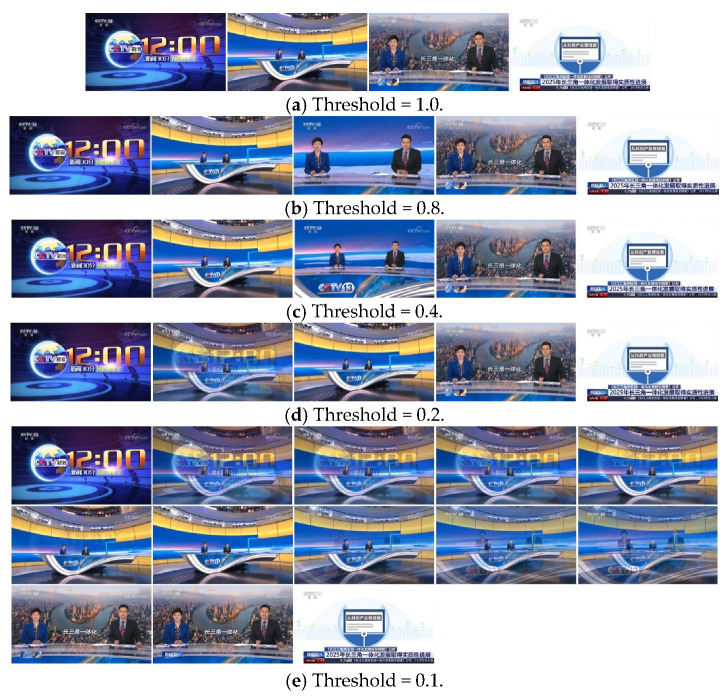
Key frame extraction results under different thresholds.

**Figure 7 entropy-23-00982-f007:**
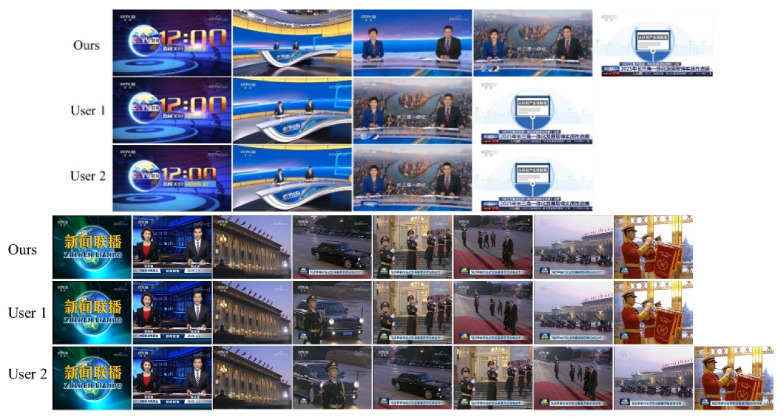
Comparison with summarized videos made manually by the users.

**Figure 8 entropy-23-00982-f008:**
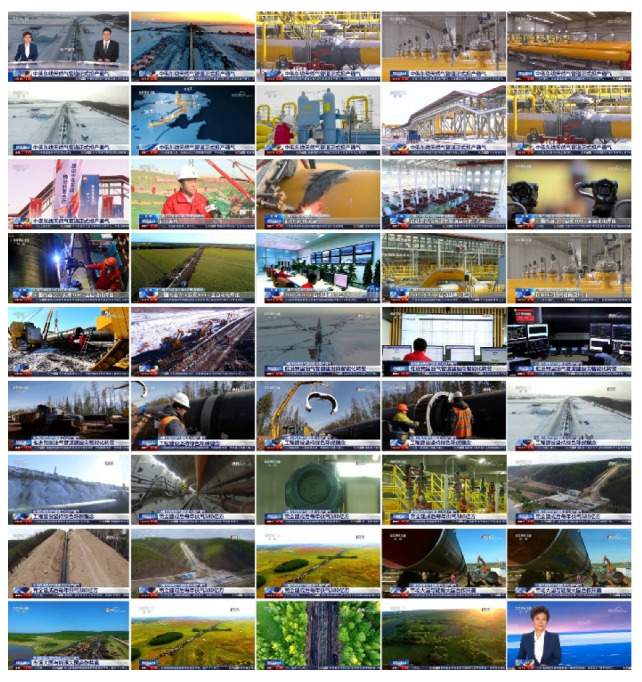
Results of summarized news videos by our method.

**Figure 9 entropy-23-00982-f009:**
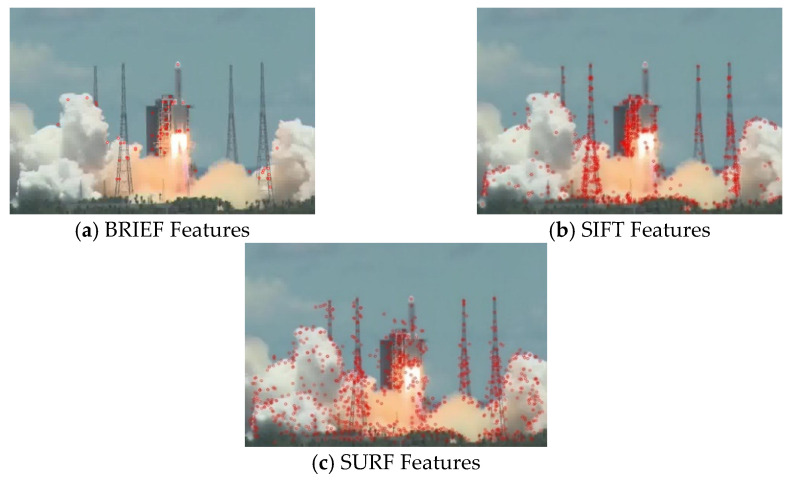
Comparison of results of different local descriptors.

**Table 1 entropy-23-00982-t001:** The details of the datasets.

Types	Number of Clips	Frames	Shots
News in 30 Minutes	6	17,675	146
CCTV News	4	8225	65
Sports Express	3	7500	76
TVSum 50	50	352,305	1720

**Table 2 entropy-23-00982-t002:** Running efficiency of our method.

Shot Boundary Detection		Key Frame Extraction
**Feature Extraction**	**Feature Matching**	1037.17 frames/s
41.27 frames/s	29.69 frames/s	

**Table 3 entropy-23-00982-t003:** Comparison of running efficiency.

Methods	Frames	Feature Extraction Speed
Ours	1350	41.27 frames/s
Li, S. [28]	1922	31.84 frames/s
Li, F. [29]	1922	26.43 frames/s

**Table 4 entropy-23-00982-t004:** Results of shot boundary detection.

Types of Video	Recall	Precision
News in 30 Minutes	96.57%	97.24%
CCTV News	100%	94.29%
Sports Express	96.05%	85.88%
Overall	97.22%	93.33%

**Table 5 entropy-23-00982-t005:** Comparison of shot detection methods.

Methods	Self-Built Dataset		Methods	TVSum50	
	Recall	Precision		Recall	Precision
Ours	97.22%	93.33%	Ours	95.93%	87.91%
Feng [30]	86.48%	93.20%			
Wang [31]	90.41%	94.28%	TransNetV2 [20]	92.56%	92.02%
Rachida [7]	96.49%	95.87%			
TransNetV2 [20]	92.68%	98.52%			

**Table 6 entropy-23-00982-t006:** Running efficiency between different local descriptors.

Local Descriptors	BRIEF	SIFT	SURF w/o Preprocess	SURF with Preprocess
Time (s/frame)	0.01295	0.09725	0.07084	0.06230

**Table 7 entropy-23-00982-t007:** Intra-class distances between different color spaces.

Color Spaces	RGB	HSV
Average intra-class distances	0.00377	0.00335

## Data Availability

Not applicable.

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
