# Peer review of "News Video Summarization Combining SURF and Color Histogram Features"

_entropy, 2021, doi:10.3390/e23080982_

Round 1
Reviewer 1 Report
The authors present an approach for news video summarization combining SURF and color histogram features. They also proposed a method for temporal segmentation, detecting abrupt and gradual boundaries. Overall, the paper structure is adequate, even though it needs some adjustments.
- The authors use classical methods for video summarization. Almost all methods were used in: E. J. Y. C. Cahuina and G. C. Chavez, "A New Method for Static Video Summarization Using Local Descriptors and Video Temporal Segmentation," 2013 XXVI Conference on Graphics, Patterns and Images, 2013, pp. 226-233.
- Are the authors using all keypoints or the more relevant for matching?
- The authors need to do an ablation study and evaluate different local descriptors and color spaces. Is it necessary the transformation of the color space? What is the performance using RGB space?
Reviewer 2 Report
The authors addressed most of my remarks on the prior version of the document. I still consider that the actual contribution is borderline but can be accepted for publication.
Author Response
I would like to thank the respected referee for your decision.
This manuscript is a resubmission of an earlier submission. The following is a list of the peer review reports and author responses from that submission.
Round 1
Reviewer 1 Report
The authors propose a news video summarization based on SURF features and a clustering algorithm. There is no novelty in the proposal.
The authors are using well-known algorithms for shot boundary detection, video summarization, and keyframe selection.
Reviewer 2 Report
The authors propose a video summarization schema based on a combined assessment that takes into account a concordance amount of SURF features among frames and a color histogram clustering method. As such, the ideas are far from being novel, between 2005 and 2015 there were literally hundreds of papers proposing similar combinations, and after that research switched to deep learning methods.
Apart from that, I have concerns about the overall methodology and development:
0) The authors don't mention, and it is not clear in the results section, whether the summarization technique they present intend to provide a storyboard or skimming. The former appears to be the case (see remark 3 below) but this would have to be stated from the beginning.
1) Logo elimination: the authors propose a quite ineffective idea (i.e., cropping the 15% bottom and 20% top of the frame). This will certainly do away 35% of the very latent information present. It would be much more simple to do binary operations among successive frames to detect invariances related to logos and then to filter out that information. Also, Fig. 2 is certainly not particularly showing the idea, since the authors are using a frame with logos (left) and without (right) (this can be clearly noted in the top left and right part).
2) The authors mention brute-force and FLANN methods for key-point matching, while in the literature most papers use RANSAC instead, which is faster and more robust.
3) The authors use inadequately the concept of "key frame". F.e., in lines 126-127: "Then, the improved clustering algorithm is used to realize the key frame extraction in each shot." The concept of key frame, from traditional animation, is related to the frames that are beginning and ending hard or soft transitions. Extrapolated to video summarization, key frames are the beginning and ending of takes. Perhaps the authors should revise the manuscript and use a less confusing term, "take representative frame" for instance.
4) In lines 270-271 the authors speak about "correctly detected shots", and thus apply traditional precision and recall measures. However, the source of the ground truth is never explained. Do they have an access to a particular gold-standard present in the datasets. If so, that must be exposed in detail. Also, they perform a "subjective evaluation" in which they only juxtapose the detected frames with frames selected by "User1" and "User2", which clearly do not constitute any solid evidence. In these cases, more "users" and videos have to be considered, and some consensus measure (f.e., Cohen's kappa) should be considered.
5) The comparison with other methods is weak, the most prominent "pre-CNN" methods (f.e. [1, 2, 3] are not even cited, and the currently used datasets (f.e., SumMe) are not considered. It is also advisable that they consider reading the survey published in [4] to better understand the state-of-the-art in the topic.
6) I am attaching a version of the original manuscript in which I highlighted the most salient typos or grammar mistakes. I strongly recommend the authors to have their manuscript revised by someone proficient in technical English writing.
[1] Liu, D., Hua, G., Chen, T.: A hierarchical visual model for video object summarization. IEEE transactions on pattern analysis and machine intelligence 32(12) (2010) 2178–2190.
[2] Zhang, S., Zhu, Y., & Roy-Chowdhury, A. K. (2016). Context-aware surveillance video summarization. IEEE Transactions on Image Processing, 25(11), 5469-5478.
[3] Jurandy Almeida, Neucimar J. Leite, Ricardo da S. Torres, VISON: VIdeo Summarization for ONline applications, Pattern Recognition Letters,
Volume 33, Issue 4, 2012, Pages 397-409, ISSN 0167-8655,
[4] https://www.researchgate.net/publication/338526880_Digital_Video_Summarization_Techniques_A_Survey

Reviewer 3 Report
This paper proposes a new feature that combines SURF and color histogram feature together for processing videos.
The method has been compared with a state-of-the-art methods and achieve comparable or better results. The method is well presented.
The main suggestion is to improve the presentation, especially in the section of results. For example, the qualitative results such as Fig. 6 are not clear when I enlarge the figure to see the details. Also, in the quantitative results such as Table 3, please add the name of the compared methods. Thanks!